# Rapid one-step [18]F-radiolabeling of biomolecules in aqueous media by organophosphine fluoride acceptors

Huawei Hong[1], Lei Zhang [2], Fang Xie[3], Rongqiang Zhuang[1], Donglang Jiang[3], Huanhuan Liu[1], Jindian Li[1], Hongzhang Yang[1], Xianzhong Zhang [1], Liming Nie[1] & Zijing Li [1]

Currently, only a few [18]F-radiolabeling methods were conducted in aqueous media, with non-macroelement fluoride acceptors and stringent conditions required. Herein, we describe a one-step non-solvent-biased, room-temperature-driven [18]F-radiolabeling methodology based on organophosphine fluoride acceptors. The high water tolerance for this isotope-exchange-based [18]F-labeling method is attributed to the kinetic and thermodynamic preference of F/F over the OH/F substitution based on computational calculations and experimental validation. Compact [[18/19]F]di-*tert*-butyl-organofluorophosphine and its derivatives used as [18]F-labeling synthons exhibit excellent stability in vivo. The synthons are further conjugated to several biomolecular ligands such as c(RGDyk) and human serum albumin. The one-step labeled biomolecular tracers demonstrate intrinsic target imaging ability and negligible defluorination in vivo. The current method thus offers a facile and efficient [18]F-radiolabeling pathway, enabling further widespread application of [18]F.

[1] State Key Laboratory of Molecular Vaccinology and Molecular Diagnostics & Center for Molecular Imaging and Translational Medicine, School of Public Health, Xiamen University, 361102 Xiamen, Fujian, China. [2] Tianjin Engineering Technology Center of Chemical Wastewater Source Reduction and Recycling, School of Science, Tianjin Chengjian University, 300384 Tianjin, China. [3] PET center, Huashan Hospital, Fudan University, 200235 Shanghai, China. Correspondence and requests for materials should be addressed to L.N. (email: nielm@xmu.edu.cn) or to Z.L. (email: zijing.li@xmu.edu.cn)

Positron emission tomography (PET) is a highly sensitive, quantitative, noninvasive imaging technique, and therefore is one of the most popular imaging modalities for disease diagnosis and monitoring of therapeutic effects[1]. The [18]F-radionuclide contributes greatly to the success of this imaging method and is often preferred for PET imaging due to its wide availability, favorable half-life (109.8 min), high isotopic purity, and low positron energy (640 keV), allowing sequential acquisition of PET images with a relatively high resolution[2]. However, to synthesize [18]F-labeled PET tracers, traditional labeling methods are mostly based on carbon-fluorine bond formation, which usually demands multiple steps and/or harsh reaction conditions such as heating at high temperatures in organic media[3]. Moreover, most [18]F-radiolabeling reactions require strict anhydrous conditions due to the limited reactivity of the fluoride anion as a nucleophile in aqueous solution[4] (Fig. 1a). Feasible and efficient [18]F-labeling methods in physiological media remain highly desirable for the introduction of [18]F into useful biomolecules, such as water-soluble peptides and antibodies that bind to overexpressed receptors in various inflammatory tissues and tumors[2].

A few studies[5–8] reported that fluorinase enzyme can catalyze the incorporation of [18]F into PET radiotracers under aqueous conditions. However, the high substrate specificity severely limits the scope of application for this enzyme method[7]. Several direct aqueous [18]F-labeling approaches have been reported, using organofluorosilicons[2, 9–15], aluminum complexes[16–19] or tri-fluoroborates[20–28] as non-carbon fluoride acceptors to afford one-step labeling of peptides. These non-macroelement fluoride acceptor-based methods exploit their Lewis acid characteristics. Lewis acid promoters, such as $SnCl_4$ and $AlCl_3$, are sometimes required to activate the fluorine-element bond to facilitate [[18]F]F⁻ incorporation in these systems[29]. The high bond enthalpy and low activation energy for the binding of Si, Al, and B with F⁻ lead to the formation of stable bonds with [18/19]F⁻ instead of being attacked by and bonded with OH⁻[30].

However, these methods still pose obvious drawbacks: (i) limited stability and high lipophilicity of the organosilicon-based [18]F-labeling blocks; (ii) specific pH requirement for trifluoroborate-based [18]F-labeling; (iii) steric effect of bulky Al-[18]F-based chelate synthons; and (iv) potential biosafety issue due to possible metal contamination in the final product. Recently, a bulky [GaF₃(BnMe₂-tacn)], another metal chelate system, was labeled in aqueous solutions by [18]F/[19]F isotopic exchange but showing slight defluorination of the [18]F-labeled product in plasma[29]. Collectively, non-solvent-biased [18]F-labeling methods with macroelement-composed fluoride acceptors remain unexplored.

The phosphorus (P) element is essential for life and phosphines are components of deoxyribonucleic acid, ribonucleic acid, adenosine triphosphate, and phospholipids[31]. The formation heat of phosphine fluoride radicals is $-12.5 \pm 5$ kcal mol⁻¹ while those of silicon fluoride, boron fluoride, and fluorocarbon radicals are $-4.8 \pm 3$, $-27.7 \pm 3.3$, and $61.0 \pm 2$ kcal mol⁻¹, respectively[32]. Therefore, phosphorus, boron, and silicon readily form more stable products with fluorine (compared to fluorocarbon). Phosphorus is strongly fluorophilic based on previous reports[33, 34]. For example, the affinity of F-P in $PF_5$ (380 kJ mol⁻¹) is even higher than that of F-B in $BF_3$ (346 kJ mol⁻¹). Preparation of compounds containing the P-[18]F bond was demonstrated by [18]F-labeling of phospho-pesticide[35] and NCH-$PF_5$[33] in organic solvents under heating condition.

Here we rationally design and develop organophosphine precursors that allow for easy synthesis and efficient radiolabeling with high radiochemical yields (RCYs) and molar activities under mild conditions. In this way, [18]F-labeled biomolecules are successfully obtained in either organic solvents or aqueous media with high RCY up to 100% within 15 min, as shown in Fig. 1b. Proof-of-concept of [18]F-labeling with heat/solvent-sensitive biomolecules in one step is demonstrated by biomolecular tracers include [18]F-labeled human serum albumin (HSA) for blood pool imaging and [18]F-labeled c(RGDyk) which targets the integrin $\alpha_v\beta_3$ receptor of the U87MG cell (glioblastoma). Thus, the method reported here permits non-specialist-manageable rapid [18]F-labeling of extensive biomolecules and their use in PET imaging.

## Results

**Mechanism of rapid H₂O-resistant [18]F/[19]F exchange.** We first performed computational density functional theory (DFT) calculations to simulate the F/F isotope exchange processes on the organophosphine fluoride acceptors with substitutions of varying steric hindrance. Using the B3LYP method[36–40], two-step addition-elimination pathways for both F/F isotope exchange and the competing OH/F substitution (i.e., hydrolysis) were identified for

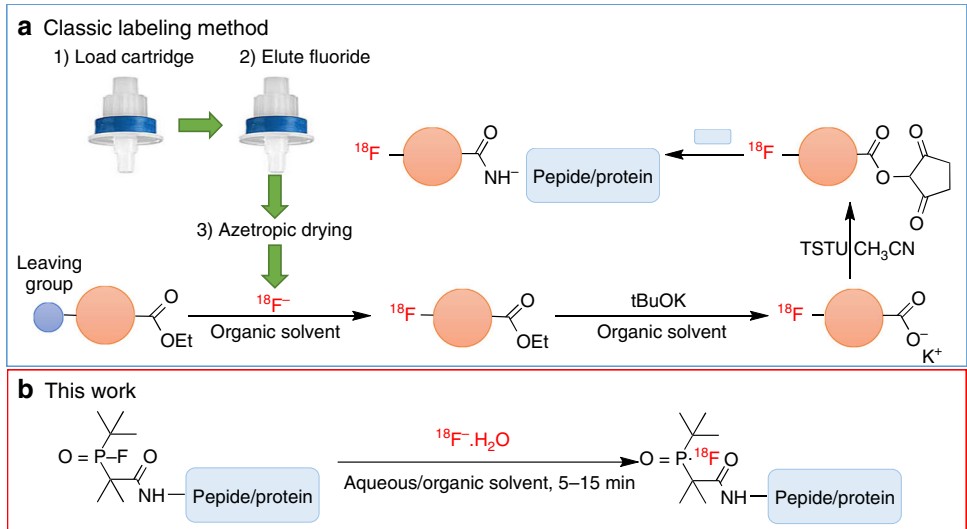

**Fig. 1** Overview of [18]F-labeling methods for solvent/heat-sensitive biomolecules. **a** Commonly used multistep synthetic route beginning with drying of aqueous [[18]F]F⁻. **b** One-step [18]F-labeling of biomolecules via an organophosphine fluoride acceptor that allows for efficient labeling with good RCYs (>50%) under mild conditions within 5–15 min

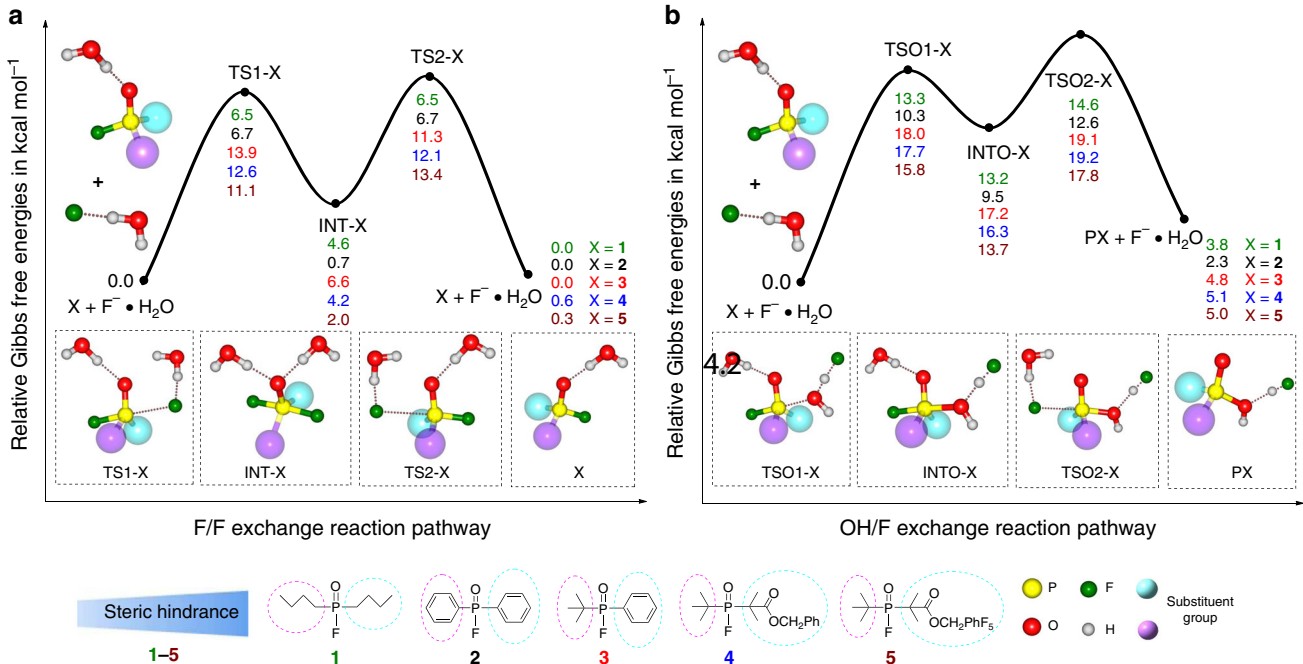

**Fig. 2** Mechanism of H$_2$O-resistant $^{18}$F/$^{19}$F exchange interpreted by free energy parameters. Geometries are optimized at the B3LYP/6-31+G* level of theory, and single point calculations are performed at the CAM-B3LYP/6-311++G** level of theory. Two substituent groups on the phosphorus center are simplified as two large spheres in the molecular structures for clarity. **a** Reaction pathways and free energy (kcal mol$^{-1}$) profiles for the F/F isotope exchange process of five selected reactant systems. **b** Reaction pathways and free energy (kcal mol$^{-1}$) profiles for the OH/F exchange process of the same five selected reactant systems

five representative organophosphine fluorides (**1**–**5**). Detailed reaction mechanisms and free energy profiles are shown in Fig. 2. Such a mechanistic path suggests an inversion of the tetravalent phosphorus after a single F/F exchange. Hence, the overall reactions are not strictly free energy neutral but negligible. The OH/F exchange processes share similar reaction pathways with the exception that [F…H$_2$O]$^-$ binds with the phosphorus center using its oxygen atom in the first addition step.

The calculated free energy profiles show that the pentacoordinate intermediates are unstable compared to the initial materials for both F/F isotope exchange and OH/F substitution. A comparison of the relative stabilities of TS1-X and TS2-X (X = **1**–**5**) to those of TSO1-X and TSO2-X shows that the free energy barriers for the F/F exchange processes were distinctly lower than the corresponding values for the OH/F substitution. The differences in relative free energy values between TS1-X and TSO1-X for the **1**–**5** reaction systems were −3.6, −4.1, −5.1, −7.2, and −4.7 kcal mol$^{-1}$, respectively, while those between TS2-X and TSO2-X for the five reaction systems were −5.9, −7.8, −7.1, −8.1, and −4.4 kcal mol$^{-1}$, respectively. These results strongly indicate that the F/F isotope exchange is kinetically more favorable than the corresponding OH/F substitution for all of the reactant systems. According to the calculated relative free energies of the transition states, the predicted rates for F/F isotope exchange follow the order of **1** ≈ **2** > **5** > **4** > **3**. In addition, the OH/F substitution processes exhibit endothermic free energy values (*e.g.*, ΔG = 2.3–5.1 kcal mol$^{-1}$), suggesting that these processes are neither spontaneous nor thermodynamically favorable. Overall, the DFT calculations demonstrate that the $^{18}$F/$^{19}$F isotope exchange occurs automatically rather than the competing OH/F substitution in terms of both kinetics and thermodynamics (Table 1).

**Synthesis of organofluorophosphine fluoride acceptors**. Next, the designed organofluorophosphine fluoride acceptors were prepared via oxidative coupling between various hydrophosphine oxide and F$^-$. Specifically, dibutylphosphinic fluoride (**1**), diphenylphosphinic fluoride (**2**), *tert*-butyl(phenyl)phosphinic fluoride (**3**) and benzyl 2-(*tert*-butylfluorophosphoryl)-2-methylpropanoate (**4**) were synthesized from phosphine oxides using the Reformatsky-type reaction[41], followed by copper-promoted fluorination of phosphine[42]. Compound **1** was found to be difficult to obtain and only $^{19}$F and $^{31}$P NMR spectra were obtained because of its volatility and low yield.

**Radiochemistry**. For radiolabeling tests, [$^{18}$F]F$^-$ solution unloaded from a cyclotron target was diluted and added to reaction vials pre-charged with the organofluorophosphine fluoride acceptors. Rapid $^{18}$F-labeling was achieved in high RCYs by $^{18}$F/$^{19}$F isotope exchange in aqueous or organic solvents without heating. The RCYs were further measured by changing H$_2$O ratio from 0 to 95% at graded precursor concentrations and reaction temperatures. For the first try, one-step $^{18}$F-labeling in >95% aqueous solutions was achieved with >50% RCY under mild conditions[31]. As the precursor concentrations and reaction temperatures increased, the RCYs improved significantly. Under the same labeling conditions, [$^{18}$F]**3** and [$^{18}$F]**4** were more rapidly labeled with higher yields than [$^{18}$F]**2**. [$^{18}$F]**3** and [$^{18}$F]**4** achieved a 100% labeling yield in dimethyl sulfoxide (DMSO) at 75 °C, as shown in Table 2. The molar activity of manually labeled [$^{18}$F]**4** was measured to be 0.01–0.05 Ci μmol$^{-1}$ (depending on the amount of starting radioactivity, decay-uncorrected) at the end of synthesis. Detailed radiolabeling procedures and calculations are described in the supporting information.

**Stability of $^{18}$F-labeled organofluorophosphine**. In vitro stabilities of [$^{18}$F]**2**, [$^{18}$F]**3**, and [$^{18}$F]**4** in human serum, saline, and ethanol, respectively, are shown in Supplementary Figure 5. Among

**Table 1 Comparison of flourophosphines with various degrees of steric hindrance**

| Compound | Natural bond orbitals charge Analysis (e) | Delocalized molecular orbital (Hartree) | F/F–OH/F (kcal mol$^{-1}$)[a] | | RCY (%)[b] | In vivo stability[c] |
|---|---|---|---|---|---|---|
| | | | TS1-TSO1 | TS2-TSO2 | | |
| **1** | P  2.227; O  -1.080; F  -0.594 | 0.0386 | -6.8 | -8.1 | --- | --- |
| **2** | P  2.255; O  -1.067; F  -0.581 | 0.0315 | -3.6 | -5.9 | 27 ± 4 | --- |
| **3** | P  2.268; O  -1.078; F  -0.588 | 0.0559 | -4.1 | -7.8 | 92 ± 5 | 40.3% |
| **4** | P  2.275; O  -1.086; F  -0.582 | 0.0488 | -5.1 | -7.1 | >97 | 100% |

Results of RCY are presented by means ± standard deviations ($n = 3$)
[a]P–F bond energy calculated by Gaussian 09
[b]All RCYs are experimental values acquired under the same conditions, where 1.0 μmol of precursor was dissolved in 100 μL of DMSO, 100 μL of [18F]F- aqueous solution from the cyclotron target was added, and the reaction was carried out at 75 °C for 15 min. All yields are presented at EOS without decay correction, and each experiment was repeated for 3 times
[c]Fraction of the intact 18F-labeled fluorophosphine synthons in the plasma of male BALB/c nude mice at 120 min after tail vein injection

**Table 2 RCYs of 18F-labeled fluorophosphine synthons under various conditions**

**2**, R$^1$ = Ph, R$^2$ = Ph; **3**, R$^1$ = Ph, R$^2$ = t-Bu; **4**, R$^1$ = t-Bu, R$^2$ =CH(CH$_2$)$_2$COOC$_6$H$_5$

| Compound | Scale[a] | Solvent (H$_2$O/DMSO)[b] | T (°C) | RCY[c] (%) |
|---|---|---|---|---|
| **2** | 3 | 0/100 | 75 | 27 ± 4 |
| **2** | 3 | 50/50 | 75 | 5 ± 4 |
| **2** | 3 | 95/5 | 75 | 0[d] |
| **3** | 3 | 0/100 | 75 | 92 ± 5 |
| **3** | 3 | 50/50 | 75 | 89 ± 4 |
| **3** | 3 | 95/5 | 75 | 93 ± 3 |
| **4** | 3 | 0/100 | RT | 93 ± 6 |
| **4** | 3 | 0/100 | 75 | >97 |
| **4** | 3 | 50/50 | RT | 80 ± 8 |
| **4** | 3 | 50/50 | 75 | >97 |
| **4** | 3 | 95/5 | RT | 50 ± 5 |
| **4** | 3 | 95/5 | 75 | 60 ± 6 |

[a]μmol
[b]Total volume: 200 μL
[c]Each reaction was performed with 2–10 mCi of [18F]F- following the Method I-VI in the Supplementary Information. Non-decay-corrected RCYs determined by radio-HPLC are presented by means ± standard deviations ($n = 3$)
[d]The precursor was not soluble

these 18F-labeled fluorophosphines, [18F]**4** exhibited the highest in vivo stability (Table 1). During a metabolic stability examination, [18F]**4** remained 100% intact at 2 h post-injection and was more stable than [18F]**3**, which is in good agreement with the predictions from the free energy barrier calculations (Table 1). MicroPET/CT imaging with [18F]**4** in healthy ICR mice was then performed to investigate the in vivo metabolic process. No increase in bone uptake was observed up to 120 min post-injection. The radioactivity was primarily concentrated in the bladder and gallbladder over time (see Supplementary Figure 7). 2,3,5,6-Tetrafluorophenyl 2-(tert-

butylfluorophosphoryl)-2-methylpropanoate (DBPOF-COOC$_6$HF$_4$, **5**), a derivative with an activated ester, was synthesized and conjugated to several peptide ligands of clinical interest[43, 44] to obtain one-step-labeling precursors.

**Radiosynthesis of 18F-DBPOF-c(RGDyk) and PET imaging.** Using a manual labeling procedure, 18F-DBPOF-c(RGDyk) as a peptide ligand was prepared at an average RCY (the amount of activity in the final product expressed as the percentage of starting

activity) of >30% in 25 min with >98% purity after purification with a C-18 solid-phase extraction cartridge. In order to obtain cGMP compliant $^{18}$F-DBPOF-c(RGDyk), we established an automated procedure using a commercial multifunction radio-synthesis module PET-MF-2V-IT-I (Beijing PET Technology, China). The schematic layout of the automated synthesis module is shown in Supplementary Figure 4.

Quality control analysis protocols were established according to the United States Pharmacopeia guidelines for PET radio-pharmaceuticals. Average RCY (the amount of activity in the final product expressed as the percentage of starting activity) was 15–25% for the automated procedure. The radiochemical purity of $^{18}$F-DBPOF-c(RGDyk) was confirmed to be >95% by radio-HPLC analysis. The molar activity was measured to be 0.06–0.13 Ci μmol$^{-1}$ for the product prepared using the automated module, compared to 0.006–0.01 Ci μmol$^{-1}$ for the manually prepared product. MicroPET/CT imaging data, shown in Fig. 3a, indicated an uptake of $^{18}$F-DBPOF-c(RGDyk) with standardized uptake value (SUV) of 0.76 in U87MG gliomablastoma, and a target to non-target ratio of 2.4. No obvious signal was detected in the tumor as illustrated in the inset image of Fig. 3a, demonstrating that the specific uptake was blocked by DBPOF-c(RGDyk).

**Radiosynthesis of $^{18}$F-DBPOF-HAS and PET imaging.** Human serum albumin (HSA, 66.5 kDa) is a heat-sensitive globular protein that exhibits high solubility and stability with a plasma half-life of 16–18 h[45]. DBPOF-HSA was prepared by simple conjugation of DBPOF-COOC$_6$HF$_4$ to HAS within 30 min. By

manual $^{18}$F-labeling, the RCY (the amount of activity in the final product expressed as the percentage of starting activity) of $^{18}$F-DBPOF-HSA was >5% with molar activity of ~0.03 Ci μmol$^{-1}$ in shorter synthesis time than other radiofluorination strategies.

PET imaging of healthy female Wistar rats was conducted 1 h after intravenous injection of $^{18}$F-DBPOF-HSA. A whole-body PET/CT of a healthy rat 1 h after $^{18}$F-DBPOF-HSA injection through tail vein is demonstrated in Fig. 3c. The heart ventricles the and the peripheral blood vessels are clearly revealed. Other organs can be also observed but with lower activity concentrations than that in the central vessels. Analysis of $^{18}$F-DBPOF-HSA biodistribution in rats showed high retention in blood with SUVs of 2.4 ± 0.6, 1.4 ± 0.7, and 1.0 ± 0.1 at 0.2, 0.5, and 1 h, respectively. The tissue uptake from the microPET/CT experiments agreed well with the biodistribution results at matched time points. These results indicated that the introduction of DBPOF synthon into HSA and subsequent radiolabeling did not affect the structural or functional integrity of HSA.

## Discussion

Biomolecules, such as proteins, carbohydrates, and nucleic acids, are excellent vectors for monitoring physiological and biochemical lesions via PET imaging due to their high affinity and high specificity for many pathogenic targets[46] as well as superior biocompatibility. Most of these vectors are soluble and remain intact in water alone or mixed aqueous solvents rather than in organic solvents, while anionic [$^{18}$F]F$^-$, with its high solvation energy, becomes inactive in the same solvents. Consequently, the

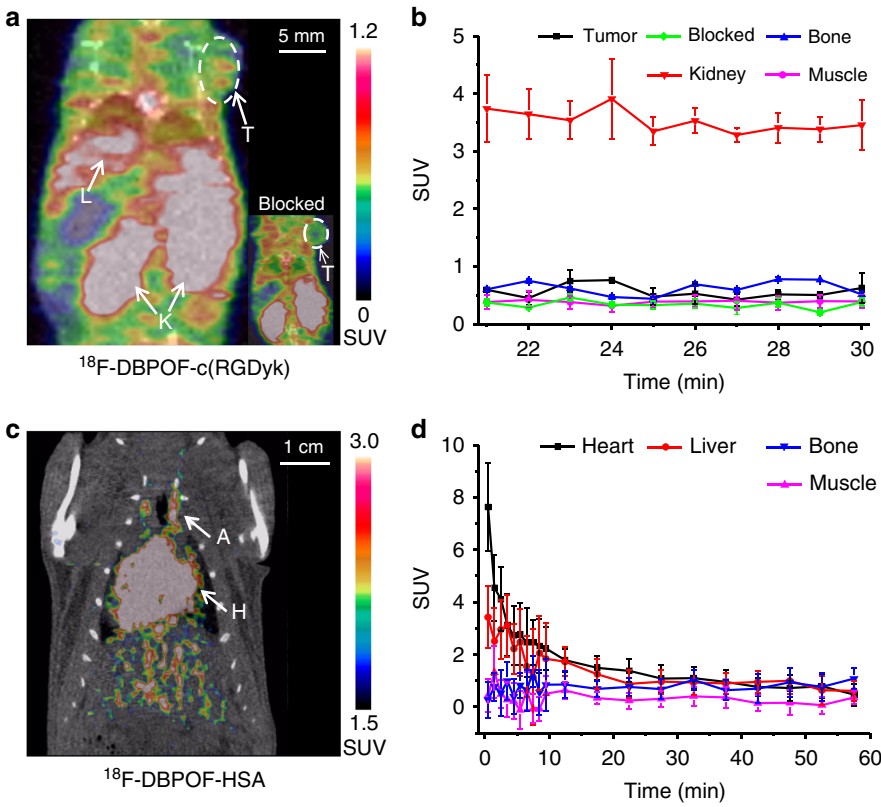

**Fig. 3** MicroPET/CT imaging with $^{18}$F-DBPOF-c(RGDyk) and $^{18}$F-DBPOF-HSA. **a** A PET image from a dynamic scan with $^{18}$F-DBPOF-c(RGDyk) in U87MG nude mice bearing gliomablastoma, reconstructed at 23 min post-injection with the tumor (T), live (L), and kidney (K) indicated by white arrows. The inset image in the right corner showed no specific tumor uptake indicated by a white arrow. **b** Time-activity curves of $^{18}$F-DBPOF-c(RGDyk) in the tumor, bone, blood, kidney and muscle. Uptake values are presented by means ± standard deviations (n = 3). **c** A PET image from a 60 min dynamic scan with $^{18}$F-DBPOF-HAS in healthy female Wistar rats, reconstructed at 60 min post-injection with the heart (H) and artery (A) indicated by white arrows. **d** Time-activity curves of $^{18}$F-DBPOF-HSA in the heart, liver, bone and muscle. Uptake values are presented by means ± standard deviations (n = 3). Source data are provided as a Source Data file

direct incorporation of $^{18}$F is challenging, especially in the case of temperature-, pH- and solvent-sensitive macromolecules. Fortunately, efficient aqueous labeling approaches are theoretically feasible by employing highly fluorophilic fluoride acceptors with very low activation energy and high hydrolysis energy for selective fluoro-substitution. Indeed, fluorophosphine fluoride acceptors have been shown to undergo rapid radiofluorination reaction in at least partially aqueous media[10].

Theoretical DFT calculations confirmed that the F/F exchange of organophosphines is more favorable both kinetically and thermodynamically than the OH/F exchange, which is consistent with our experimental findings. The detailed reaction mechanisms and free energy variations were illustrated for the organophosphine labels with five representative reactants. In our calculations, the 1:1 hydrogen-bond complex of $H_2O$ with $F^-$ ($F^- \cdot H_2O$) was modeled as a nucleophile to account for the solvation effect of water. Nucleophilic substitution at the tetravalent phosphorus center may occur via either a concerted pathway or a stepwise addition-elimination pathway. The latter method involves a pentavalent adduct intermediate owing to the addition of a nucleophile without the loss of a leaving group.

In addition to the spontaneity and aqueous tolerance of these radiolabeling reactions, the compact prothesis configuration and favorable in vivo behavior of the resulting PET tracers are also important. [$^{18}$F]**2** was found to be unstable at pH 7.4–7.6 in human serum, saline, and ethanol because the steric hindrance of benzene is relatively small. In contrast, [$^{18}$F]**3** and [$^{18}$F]**4** exhibited remarkable thermal and hydrolytic stability. The stabilities of $^{18}$F-labeled fluorodiphenylphosphines are consistent with the steric parameters of these compounds, i.e., greater steric hindrance leads to better stability. By virtue of their small size, zero net charge, and negligible decomposition, the physicochemical properties of the $^{18}$F-labeled tracers prepared via the organophosphine fluoride acceptors are similar to those of the unlabeled analogs.

Phosphorus compounds are used in many modern homogeneous catalyst systems because the P-X bonds are thermally robust in general and unlikely to break down by transition metals at the catalysis conditions[47]. Fluorophosphines with bulky or electron-withdrawing substituents, such as $(C_6F_5)_2PF$, $(CF_3)_2PF$, and $tBu_2PF$, are also found to be thermally stable[48–51]. Synthesis and screening of more fluorophosphine fluoride acceptors with various leaving groups are continuing in our laboratories.

In summary, we have developed a procedure for the radiolabeling of a series of organofluorophosphine biomolecules using $^{18}$F-fluoride. Low activation energy is required for the formation of these P-F bonds in comparison to P-OH bonds. As a result, $^{18}$F-labeling can be performed in aqueous media and high radiochemical yields achieved even at room temperature, and acceptable molar activities that are critical for targeted imaging[52]. This inexpensive, rapid, one-step $^{18}$F-radiolabeling procedure holds great promise for widespread applications in radiopharmaceutical industry to produce a variety of biomolecules as potential PET tracers.

## Methods

**Automated radiosynthesis of $^{18}$F-DBPOF-c(RGDyk)**. No-carrier-added [$^{18}$F]F$^-$ was obtained by bombardment of a $H_2^{18}O$ target with 11-MeV protons in a RDS 111 cyclotron (Siemens, Germany) and delivered to the PET-MF-2V-IT-I synthesis module. The aqueous [$^{18}$F]F$^-$ solution (600–1200 mCi) was mixed with a solution of the unlabeled precursor DBPOF-c(RGDyk) (0.8–2.4 mg, 1–3 μmol) in ~0.5 mL of DMSO/$H_2O$ (1:1, v/v, pH = 7.4) in the reaction vial. The vial was kept at room temperature for 15 min without stirring. The reaction mixture was diluted with 10.0 mL of $H_2O$ and loaded onto a C18 light cartridge (Waters, USA). The cartridge was flushed twice with 10.0 mL of pure water each to remove the unreacted [$^{18}$F]F$^-$, then eluted with 1.0 mL ethanol to recover 90–150 mCi of radiochemically pure product collected in a glass vial. This solution was then formulated in 0.9% sodium chloride (9.0 mL) and filtered through a 0.22-μm sterile Millex® GV

(vented filter, Millipore, Billerica, MA, USA) into a sterile dose vial (Mallinckrodt, Hazelwood, MO, USA) fitted with a vent needle (Sartorius Stedim Biotech GmbH, Göttingen, Germany). A small sample was removed for quality control analysis. The radiolabeling procedure was repeated at least three times. Detailed HPLC profiles and procedure for molar activity measurements are presented in the Supplementary Information.

**Radiosynthesis of $^{18}$F-DBPOF-HAS**. DBPOF-HAS (2–3 mg) was dissolved in 150 μL of 0.01 M phosphate buffered saline (pH = 7.36) in a glass vial. Aqueous [$^{18}$F]F$^-$ solution (50 μL, 30–50 mCi) unloaded from the cyclotron target was added to the reaction vial. The vial was gently shaken at room temperature for 15 min. With the reaction impurities (i.e., free [$^{18}$F]F$^-$) removed by a radio-HPLC equipped with a Xtimate SEC-300 column (Welch, China), 1.5–2.8 mCi of the desired product was collected. A small sample was removed for quality control analysis. Radiochemical purity was measured by a radio-HPLC with a Xtimate SEC-300 column. The radiolabeling was repeated at least three times. Detailed analytical HPLC procedures are presented in the Supplementary Information.

**MicroPET/CT imaging**. All animal studies were performed under the animal use and care regulations approved by Center of Animal Care and Use Committee, Xiamen University. Dynamic microPET-CT scans were acquired by an Inveon microPET/CT (Siemens, Germany). Reconstruction of PET images was performed with 3D OPMAP algorithm.

**Reporting summary**. Further information on experimental design is available in the Nature Research Reporting Summary linked to this article.

## Data availability
The authors declare that the data supporting the findings of this study are available within the article and its Supplementary Information Files or from the corresponding author on reasonable request.

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

## Acknowledgements

This study was supported by the National Science Foundation of China (81501534&91859113), the Excellent Youth Foundation of Fujian Scientific Committee (2018J06024), Fundamental Research Funds for the Central Universities (20720180050&20720170065), Fourth Round Fujian Health Education Joint Research Projects (WKJ2016-2-08), Scientific Research Foundation of State Key Laboratory of Molecular Vaccinology and Molecular Diagnostics (2016ZY002), and Natural Science Foundation of Fujian Province, China (2016J05200).

## Author contributions

Z.L. and L.N. conceived the study and coordinated all the research. H.H. synthesized, characterized and radiolabeled all organophosphine fluoride acceptors and performed microPET/CT imaging. Z.L., H.H., and L.N. analyzed the results and wrote the manuscript. Mechanism calculation was carried out by L.Z., F.X., D.J., R.Z., H.Y., H.L., and J.L. helped in biological and imaging experiments. D.J. and X.Z. helped in discussions of the research.
