## [Peer Review File · Nature Communications]

Reviewers' comments:

Reviewer #1 (Remarks to the Author):

This is a highly important and interesting piece that I am enthusiastic about seeing in press however there are significant problems that must be addressed prior to publication.

1) Table 2: give what R1 and R2 are in each of the cpds or clarify the structures above the table' units of concentration [a] are unclear – umol is not a concentration; [d] is not footnoted anywhere in the table.

2) The images show essentially no tumor uptake. While this clearly shows metabolic stability of the P-F bond, it's unclear why there is no apparent tumor uptake (I would caution that an uptake value of 0.15% ID/g, which is no different from blood or bone is not a significant uptake value. It would be better if the authors could show real uptake. The heat-mapped color image shows essentially no contrast – hence if uptake values were low but T:NT ratios were high, I would support the argument that this can be used for imaging tumors. However, this lack of uptake or contrast erodes the significance and utility of this work in its current form. (typically a blocking study is done to prove specific uptake but with such poor uptake values and apparent contrast ratios, a blocking study will not provide the needed insight). As it stands, it seems that there is no tumor uptake whatsoever.

3) The supplementary Scheme 1 requires rewriting – the reagents a-d do not correspond for the most part. In addition, I am incredulous that the formation of cpd 4 is possible as written – there must be an aqueous work up.

4) More discussion is required to address how CuCl₂ and CsF give the phosphonyl fluoride – is Cu(II) reduced to Cu(0) or Cu(I)? If so, this can be discussed.

5) Supplementary Figure 2 suggests in vivo metabolic stability but requires greater description – is this urinalysis, blood analysis or in vitro serum stability analysis?

6) Scale bar is required on Supplementary Figure 3: please give a number, not a value of "high" or "low". There seems to be more significant bone uptake (see spinal uptake) in this figure – the authors should comment why spine is seen.

7) Radiosynthesis figures: S16-19 show a considerable amount of material that elutes prior to the main peak or at least that the peaks are not characteristic of a pure cpd. What is the explanation for these strange peaks and/o can the authors account for this?

8) The section on Radiosynthesis of ¹⁸F-labeled organofluorophosphine fluoride acceptors in the supporting info lacks detail, and, in particular to what part of the paper this synthesis applies. There is another radiosynthesis that is given in the paper using 400-1000 Ci but this section uses 2-10 mCi. In addition, this section refers to the use of 1 mg (~2 umol) for labeling with <10 mCi. How is this relevant to clinical labeling as molar activities will be extremely low? Further to this point, various method optimizations are given I-V in the supporting info. All use 1 mg and 2-10 mCi. This would necessarily give molar activities < 0.005 Ci/umol. How do the authors account for molar activities of 5.6Ci/umol.

9) The comment that the isotope exchange is not necessary energy neutral is dubious and requires more discussion. Unless I have missed something, there is no change in bond-order and there is no stereochemical argument to be made since all substrates contain only one stereogenic atom (P) and are prepared as racemates. I would submit that the equilibrium isotope effect between F19 and F18 are very low. Hence this assertion requires more clarification to say the least.

10) It is also unclear why F-F isotope exchange should be more kinetically favored than F-OH exchange and greater discussion is warranted.

Overall I found this piece to be generally well-presented and highly important to the field of radiochemistry however significant revisions are required to address the points above. Once address, I would be willing to read a revised version.

Reviewer #2 (Remarks to the Author):

The Manuscript „Rapid One-Step ¹⁸F-Labeling of Biomolecules in Aqueous Media by Organophosphine Fluoride Acceptors“ by Zijng Li et al. describes a new and fully developed method for the production of PET-Radiotracer based on non-conventional P-¹⁸F chemistry. The group describes the synthesis and both in vitro and in vivo evaluation of a set potential precursors and radioligands. Due to the very promising results as well as the extensive evaluations performed, the described technique may broaden the spectrum of available radiotracers significantly in the future.

As radiosyntheses have to be performed in an easy, convenient, efficient as well as reliable way, new techniques such as the described P-¹⁸F chemistry are able to also significantly contribute to the availability of radiotracers as the syntheses are usually the bottleneck in tracer availability. As PET is an indispensable imaging technique to detect functional changes of tissues in vivo and plan and monitor the success of targeted therapies, the findings are of broad interest not only for the radiopharmaceutical chemistry community but also for clinical sciences and fundamental preclinical research.

As Studenov and coworkers published the first approach in P-¹⁸F chemistry (J Label Compd Radiopharm 2005; 48: 497–500), the manuscript has to be cited and the results have to be mentioned as preliminary work of other groups in the text.

The specific or better molar activities (cf. Consensus Paper: Nucl Med Biol. 2017 Dec;55:v-xi.) are crucial when performing radiosyntheses for target-specific PET imaging. The results should be therefore included in the main text and not only in the SI. Especially because of the mentioned high lipophilicity of the P-F synthons (line 161) in the main manuscript.

As ¹⁸F-labeling directly from the target water can never be performed in a cGMP/GMP compliant way (and the experimental results suggest that the labelling can only performed with target water with high to very high activity per volume), the automatized process described by the authors in the SI (including the QMA purification) should be discussed in the main text as this is the key for a potential later use of this technique in a clinical setting.

Further minor points:

89-90: c(RGDyK)-peptides are usually described to bind to Integrin $\alpha v \beta 3$ in general and in particular on U87 glioblastoma cells.

123-127, table 2, figure 2: inconsistent numbering of the compounds. Change R1-R5 to standard numbering (1, 2, 3,...).

Table 1: numbering of the compounds is missing

Table 2: cluster the results of the evaluated compounds by e.g. different background grey levels to make it more intuitive

Inconsistent citation style in references

Reviewer #3 (Remarks to the Author):

The authors have identified an interesting new chemistry that may find application in PET labelling of peptides and other small biomolecules with appropriate PK properties to be amenable to labelling with fluorine-¹⁸.

Unfortunately, the work appears rather rushed with multiple errors in figures, graphs and the text at large. While the yields are relatively high, quoted molar activity appears unrealistic in relation to the chemistry utilised and is more likely than not a result of experimental error.

It is unfortunate that the authors did not decide to pursue biomolecules of greater biological and pharmacological relevance.

The conclusion that little defluorination is observed is not supported by the data presented.

There are experimental inconsistencies between the text provided in the main paper and the supplementary information, both in experimental description and figures/graphs.

Reviewers' comments

Reviewer #1 (Remarks to the Author)

This is a highly important and interesting piece that I am enthusiastic about seeing in press however there are significant problems that must be addressed prior to publication.

Re: We highly appreciate the Reviewer's positive comments. We have improved the manuscript significantly.

1) Table 2: give what R1 and R2 are in each of the cpds or clarify the structures above the table; units of concentration^[a] are unclear— μmol is not a concentration; [d] is not footnoted anywhere in the table.

Re: Thank the Reviewer for the constructive suggestions. The structural formula of R¹ and R² have been illustrated in the new Table 2 to clarify the structures of compounds **2-4**. Furthermore, "R1-R5" in both Figure 2 and Table 2 have been replaced with their compound codes (corresponding to the new **1-5** in Scheme S1).

The concentration column header has been corrected to "scale (μmol)" to indicate the amounts of the precursors. [d] has been footnoted in the right of third line in the new Table 2.

2) The images show essentially no tumor uptake. While this clearly shows metabolic stability of the P-F bond, it's unclear why there is no apparent tumor uptake (I would caution that an uptake value of 0.15% ID/g, which is no different from blood or bone is not a significant uptake value. It would be better if the authors could show real uptake. The heat-mapped color image shows essentially no contrast – hence if uptake values were low but T:NT ratios were high, I would support the argument that this can be used for imaging tumors. However, this lack of uptake or contrast erodes the significance and utility of this work in its current form. (typically a blocking study is done to prove specific uptake but with such poor uptake values and apparent contrast ratios, a blocking study will not provide the needed insight). As it stands, it seems that there is no tumor uptake whatsoever.

Re: This point is well taken. We re-did the tumor uptake experiment with results updated in the new Figure 3. In previous experiment, the molar activity was much less than 0.1 Ci/ μmol . Using higher starting activity, the molar activity has been increased to 0.06-0.13 Ci/ μmol . The new microPET/CT data indicated an uptake of [¹⁸F]DBPOF-c(RGDyk) (SUV = 0.76) in U87MG gliomablastoma, with a target to non-target ratio of 2.4. Corresponding descriptions have been updated in Page 6 in the Result Section of the revised manuscript.

3) The supplementary Scheme 1 requires rewriting – the reagents a-d do not correspond for the most part. In addition, I am incredulous that the formation of cpd 4 is possible as written – there must be an aqueous work up.

Re: Agree. The supplementary Scheme 1 has been rewritten with the reagents a-d corresponding well to the synthetic procedures. Small organofluorophosphine synthons in the sequence of increasing steric hindrance have been recoded from **1** to **5**. di-n-Butyl-fluorophosphine (**1**), diphenyl-fluorophosphine (**2**), tert-butyl-phenyl-fluorophosphine (**3**) and di-tert-butyl fluorophosphine derivatives (**4, 5**) were synthesized.

An aqueous work up is indeed necessary for the formation of “Cpd **4**” recoded as **7** in the revised manuscript. Specifically, after the reaction was quenched with H₂O, the organic layer was separated and the water phase was extracted with ethyl acetate (3 × 50 mL) in the new Supporting Information.

Details showing how a-d correspond to syntheses are as follows:

The reagents and condition “a (tetrahydrofuran, -80 ° C → RT, 5 h)” correspond to synthesis of **6**.

The reagents and condition “b (CsF, CuCl₂, acetone, RT, 10 h)” correspond to synthesis of **1, 2, 3, 4** and **5**.

The reagents and condition “c (tetrahydrofuran, -80 ° C → RT, overnight)” correspond to synthesis of **7**.

The reagents and condition “d (Zn, tetrahydrofuran, RT, 10 h)” correspond to synthesis of **8**.

The reagents and condition “e (H₂, Pd/C, MeOH, RT, 12 h)” correspond to synthesis of **9**.

The reagents and condition “f (DCC, DMAP, 2,3,5,6-tetrafluorophenol, tetrahydrofuran, RT, 12 h)” correspond to synthesis of **10**.

4) More discussion is required to address how CuCl₂ and CsF give the phosphonyl fluoride – is Cu(II) reduced to Cu(0) or Cu(I)? If so, this can be discussed.

Re: Thank the Reviewer for this comment. How CuCl₂ and CsF give the phosphonyl fluoride follows Putative reaction mechanism, which is depicted in the following Scheme A and discussed in the Supporting Information (Page 68).

Although this reaction was performed in a single procedure, two successive steps were involved. In the first step, hydrogen of dialkylphosphites was substituted by chlorine of CuCl in an electrophilic displacement reaction with CuCl₂[Cu(II)] converted to CuCl[Cu(I)]. In the second step, the chloride of dialkyl chlorophosphate was nucleophilically displaced by fluoride with CsF converted to CsCl as described in the following reference.

Scheme A. Reaction mechanism for the synthesis of diethyl fluorophosphates.

Ref: Purohit, A. K.; Pardasani, D.; Kumar, A.; Goud, D. R.; Jain, R.; Dubey, D. K., A single-step one pot synthesis of dialkyl fluorophosphates from dialkylphosphites. Tetrahedron Letters 2015, 56(31), 4593-4595.

5) Supplementary Figure 2 suggests *in vivo* metabolic stability but requires greater description – is this urinalysis, blood analysis or *in vitro* serum stability analysis?

Re: Thank you for pointing this out. Supplementary Figure 2 is about blood analysis for stability and metabolic rates.

Experimental procedures have been described in the new Supplementary Figure 2 caption as follows:

Normal kunming mice were intravenously injected with [¹⁸F]3 or [¹⁸F]4 (2 mCi in 100 μL saline), respectively. The animals were sacrificed 2 h after injection. Blood was collected and treated with acetonitrile to precipitate insoluble proteins from the solution. The blood sample was immediately centrifuged for 5 min at 12000 rpm. The supernatants were collected and passed through a 0.22 μm Millipore filter. Then 20 μL of the supernatants were analyzed by a radio-HPLC.

6) Scale bar is required on Supplementary Figure 3: please give a number, not a value of “high” or “low”. There seems to be more significant bone uptake (see spinal uptake) in this figure – the authors should comment why spine is seen.

Re: We have assigned specific SUV values from 0 to 5 on the scale bar in the new Supplementary Figure 3. The new scale bar was set from zero to reflect the distribution of all signals clearly. The low bone uptake was attributed to ~95% RCP of [¹⁸F]4. No increase of the bone uptake was found in the new Supplementary Figure 3.

7) Radiosynthesis figures: S16-19 show a considerable amount of material that elutes prior to the main peak or at least that the peaks are not characteristic of a pure cpd. What is the explanation for these strange peaks and/o can the authors account for this?

Re: Thank the Reviewer for constructive comments. The strange peaks came from precursor impurity after long time storage and transportation. We have re-purified the precursors and re-did HPLC with clean peaks shown in the new Supplementary Figure 13-18.

8) The section on Radiosynthesis of ^{18}F -labeled organofluorophosphine fluoride acceptors in the supporting info lacks detail, and, in particular to what part of the paper this synthesis applies. There is another radiosynthesis that is given in the paper using 400-1000 mCi but this section uses 2-10 mCi. In addition, this section refers to the use of 1 mg (~2 μmol) for labeling with <10 mCi. How is this relevant to clinical labeling as molar activities will be extremely low? Further to this point, various method optimizations are given I-V in the supporting info. All use 1 mg and 2-10 mCi. This would necessarily give molar activities < 0.005 Ci/ μmol . How do the authors account for molar activities of 5.6 Ci/ μmol .

Re: Thank the Reviewer for the constructive comments. In the new manuscript, we have distinguished the radiosynthesis procedures and corresponding molar activities between manual and automatic labeling, whose details have been added in Page 74 of Supporting Information.

(1) For manual labeling of $[^{18}\text{F}]\mathbf{4}$, we used 2-10 mCi $[^{18}\text{F}]\text{F}^-$ to study radiochemical yields (RCYs, determined by radio-HPLC analysis of the crude product) under different labeling conditions. The corresponding molar activities were relatively low, 0.01-0.05 Ci/ μmol .

(2) For automatic radiosynthesis of $[^{18}\text{F}]\text{DBPOF-c(RGDyK)}$, we used 600-1200 mCi $[^{18}\text{F}]\text{F}^-$ for higher molar activity in targeted imaging. The corresponding molar activities were calculated to be 0.06-0.13 Ci/ μmol .

(3) For manual labeling of $[^{18}\text{F}]\text{DBPOF-c(RGDyK)}$, the corresponding molar activities were calculated to be 0.006-0.01 Ci/ μmol .

9) The comment that the isotope exchange is not necessary energy neutral is dubious and requires more discussion. Unless I have missed something, there is no change in bond-order and there is no stereochemical argument to be made since all substrates contain only one stereogenic atom (P) and are prepared as racemates. I would submit that the equilibrium isotope effect between ^{19}F and ^{18}F are very low. Hence this assertion requires more clarification to say the least.

Re: Since the organophosphine conformation would be changed upon inversion of the tetravalent phosphorus during F/F exchange process, the overall reaction may not be strictly

free-energy neutral ($\Delta G = 0.0-0.6$ kcal/mol) but negligible. The rapid $^{18}\text{F}/^{19}\text{F}$ isotope exchange labeling was fully proved by experiments. Detailed reaction mechanism and free energy profiles have been described in the new Figure 2 and Page 4 of new manuscript.

10) It is also unclear why F-F isotope exchange should be more kinetically favored than F-OH exchange and greater discussion is warranted.

Re: The kinetics and pathways for the reaction are determined by free-energy calculations. The free energy barriers of the F/F exchange processes are distinctly lower than those of the OH/F exchange processes. The detailed discussion have been updated in the revised manuscript (Page 4).

Overall I found this piece to be generally well-presented and highly important to the field of radiochemistry however significant revisions are required to address the points above. Once address, I would be willing to read a revised version.

Re: We would like to express our sincere gratitude to the Reviewer again. We hope the Reviewer would approve the revised version.

Reviewer #2 (Remarks to the Author)

The Manuscript “Rapid One-Step ^{18}F -Labeling of Biomolecules in Aqueous Media by Organophosphine Fluoride Acceptors” by Zijing Li et al. describes a new and fully developed method for the production of PET-Radiotracer based on non-conventional P- ^{18}F chemistry. The group describes the synthesis and both in vitro and in vivo evaluation of a set potential precursors and radioligands. Due to the very promising results as well as the extensive evaluations performed, the described technique may broaden the spectrum of available radiotracers significantly in the future. As radiosyntheses have to be performed in an easy, convenient, efficient as well as reliable way, new techniques such as the described P- ^{18}F chemistry are able to also significantly contribute to the availability of radiotracers as the syntheses are usually the bottleneck in tracer availability. As PET is an indispensable imaging technique to detect functional changes of tissues in vivo and plan and monitor the success of targeted therapies, the findings are of broad interest not only for the radiopharmaceutical chemistry community but also for clinical sciences and fundamental preclinical research.

Re: We greatly appreciate the Reviewer’s compliments.

1) As Studenov and coworkers published the first approach in P-18F chemistry (J Label Compd Radiopharm 2005; 48: 497–500), the manuscript has to be cited and the results have to be mentioned as preliminary work of other groups in the text.

Re: This literature has been cited as Reference 36 and mentioned in the introduction section (Page 3).

2) The specific or better molar activities (cf. Consensus Paper: Nucl Med Biol. 2017 Dec;55:v-xi.) are crucial when performing radiosyntheses for target-specific PET imaging. The results should be therefore included in the main text and not only in the SI. Especially because of the mentioned high lipophilicity of the P-F synthons (line 161) in the main manuscript.

Re: Thank the Reviewer for the comment. We have moved relevant descriptions from SI to Page 6 and 9 of the main text with the literature cited as Reference 53 in the updated manuscript.

3) As ¹⁸F-labeling directly from the target water can never be performed in a cGMP/GMP compliant way (and the experimental results suggest that the labelling can only be performed with target water with high to very high activity per volume), the automatized process described by the authors in the SI (including the QMA purification) should be discussed in the main text as this is the key for a potential later use of this technique in a clinical setting.

Re: We have developed another method for GMP compliant which has been added in Page 6 of the revised manuscript.

“In order to obtain cGMP compliant [¹⁸F]DBPOF-c(RGDyk), we have established an automated procedure using a commercial Beijing Patel Fluoro multifunction radiosynthesis module-2V-IT-I (PET-MF-2V-IT-I). The schematic layout of the automated synthesis module is shown in Supplementary Scheme 4. Quality control analysis protocols were established according to the United States Pharmacopeia guidelines for PET radiopharmaceuticals.”

Further minor points:

4) 89-90: c(RGDyk)-peptides are usually described to bind to Integrin $\alpha_v\beta_3$ in general and in particular on U87 glioblastoma cells.

Re: We have added the description in the revised manuscript.

5) 123-127, table 2, figure 2: inconsistent numbering of the compounds. Change R1-R5 to standard numbering (1, 2, 3,...).

Re: Thanks for the correction. We have re-numbered the compounds in the revised manuscript.

6) Table 1: numbering of the compounds is missing

Re: We have numbered the compounds in the new Table 1.

7) Table 2: cluster the results of the evaluated compounds by e.g. different background grey levels to make it more intuitive.

Re: We have clustered the results by different grey levels in the new Table 2.

8) Inconsistent citation style in references.

Re: We have formatted the citation style thoroughly in the revised manuscript.

Reviewer #3 (Remarks to the Author)

1) The authors have identified an interesting new chemistry that may find application in PET labelling of peptides and other small biomolecules with appropriate PK properties to be amenable to labelling with fluorine-18.

Re: We appreciate the Reviewer's positive comments.

2) Unfortunately, the work appears rather rushed with multiple errors in figures, graphs and the text at large. While the yields are relatively high, quoted molar activity appears unrealistic in relation to the chemistry utilized and is more likely than not a result of experimental error.

Re: We have re-done the experiments with new results presented in the revised manuscript. In the old version, we overestimated the molar activity by calculation. We have measured the molar activity by experiment, which has been updated in the new version.

3) It is unfortunate that the authors did not decide to pursue biomolecules of greater biological and pharmacological relevance.

Re: We have selected two typical biomolecules covering different biological applications for proof-of-concept validation of this feasible ^{18}F labeling method. Specifically, tumor (U87MG) angiogenesis targeting peptide represented small biomolecule agents and another blood pool mapping protein represents large biomolecule agents. We believe our method is helpful and insightful to relevant biological and pharmacological study. Our next step work is to apply our method to more disease models.

4) The conclusion that little defluorination is observed is not supported by the data presented.

Re: As Reviewer #1 also stated in Comment 6, we have assigned specific SUV values from 0 to 5 on the scale bar in the new Supplementary Figure 3. The new scale bar was set from zero to reflect the distribution of all signals clearly. The low bone uptake was attributed to ~95% RCP of [¹⁸F]4. No increase of the bone uptake was found in the new Supplementary Figure 3.

5) There are experimental inconsistencies between the text provided in the main paper and the supplementary information, both in experimental description and figures/graphs.

Re: We have double checked the experiments and data to make sure they are consistent thoroughly.

Many thanks again to the Reviewers for reviewing the manuscript.

Reviewers' comments:

Reviewer #2 (Remarks to the Author):

My comments of the original version of the manuscript entitled "Rapid One-Step ^{18}F -Radiolabeling of Biomolecules in Aqueous Media by Organophosphine Fluoride Acceptors" by Dr Li and colleagues were carefully addressed in the revised manuscript.

Due to the new promising results the results presented in the manuscript will broaden the spectrum of ^{18}F radiolabeling methods and can therefore be recommended for publication.

Reviewer #3 (Remarks to the Author):

The authors have made significant effort to correct the original document. While the methodology is not without merit, the poor molar activity and low isolated radiochemical yields limit its utility. For future reference, while it is target density dependent, as a general rule of thumb in order to meet tracer conditions molar activities must be at least $1\text{Ci}/\mu\text{mol}$ ($37\text{GBq}/\mu\text{mol}$). At the level of current molar activities, tracer conditions are not met and significant (complete) self blocking will occur. Homologous blocking experiments will confirm that the observed T:NT ratio of 2.4 is entirely the result of either blood volume or differences in non-specific binding.

The low isolated radiochemical yields are surprising and some way off from the observed radiochemical conversions observed using HPLC analysis of reaction aliquots. It is unclear what may underlay this discrepancy but as things stand, the chemistry is no more efficient than existing methods.

While the idea is novel, the low molar activity and modest isolated yields limit the use of the presented methodology to very specific applications only.

Reviewers' comments

1) The authors have made significant effort to correct the original document. While the methodology is not without merit, the poor molar activity and low isolated radiochemical yields limit its utility. For future reference, while it is target density dependent, as a general rule of thumb in order to meet tracer conditions molar activities must be at least 1 Ci/ μmol (37 GBq/ μmol). At the level of current molar activities, tracer conditions are not met and significant (complete) self blocking will occur. Homologous blocking experiments will confirm that the observed T:NT ratio of 2.4 is entirely the result of either blood volume or differences in non-specific binding.

Re: This point is well taken. We agree that high molar activity is critical for specific targeted imaging. In our case of isotope exchange labeling method, the molar activity can be improved by increasing starting molar activity of [^{18}F]F $^-$ and/or decreasing molar amount of the precursor. Consequently, the molar activity can be increased to 1 Ci/ μmol or even higher. Homologous blocking experiment has been conducted and illustrated in the new Fig. 3 to prove the targeting ability. No obvious signal was detected in the tumor as shown in the inset image of Fig. 3a, demonstrating that the specific uptake was blocked by DBPOF-c(RGDyK). Thus T:NT (2.4) was not contributed by blood or non-specific binding but targeting ability. Moreover, the targeting ability of RGD monomer is relatively limited. More effective ligands with better targeting ability will be labeled with optimized molar amount and procedures in the near future.

2) The low isolated radiochemical yields are surprising and some way off from the observed radiochemical conversions observed using HPLC analysis of reaction aliquots. It is unclear what may underlay this discrepancy but as things stand, the chemistry is no more efficient than existing methods.

Re: We appreciate the Reviewer's suggestion. We have done new experiments and obtained higher radiochemical yields as follows. The RCYs (the amount of activity in the final product expressed as the percentage of starting activity) of ^{18}F -DBPOF-c(RGDyK) were >30% (by manual labeling) and 15-25% (by automatic labeling) as well as the RCY of ^{18}F -DBPOF-HSA was greater than 5%.

We consider it reasonable because the conjugated targeting biomolecules vary from size and structures, especially, containing some active group to impact the F $^-$ attacking, the RCY decrease accordingly compared with labeling the organophosphine synthons alone.

3) While the idea is novel, the low molar activity and modest isolated yields limit the use of the presented methodology to very specific applications only.

Re: We have addressed this issue in the Responses to Comments 1 and 2 with contents updated in the revised manuscript.

REVIEWERS' COMMENTS:

Reviewer #2 (Remarks to the Author):

The authors performed additional experiments and could demonstrate a radiochemical yield of >30% for manual labelling and 15-25% using an automated labeling protocol. The protein HSA could be labelled in yields of more than 5%. In addition, the authors demonstrated the specificity of the tumor-uptake of ^{18}F -DBPOF-c(RGDyk) by blocking experiments. Hence, the principal applicability of the described new labelling method was demonstrated.

For other targets, a higher molar activity will be necessary but this was neither in the focus of this fundamental work nor does it limit the merit of this very exciting manuscript, significantly enriching the palette of ^{18}F -radiolabeling reactions with high future potential.